


# Synoptic-intraseasonal variability control on high chlorophyll-a events in the Puyuhuapi Fjord, Chilean Patagonia.

Reynier Bada-Diaz[1,2,3], Martin Jacques-Coper[2,3,5], Laura Farías[1,3,6], Diego Narváez[1,5], Italo Masotti[3,4]

[1]Departamento de Oceanografía, Universidad de Concepción, Concepción, CP 4030000, Chile
[2]Departamento de Geofísica, Universidad de Concepción, Concepción, CP 4030000, Chile
[3]Centro de Ciencia del Clima y la Resiliencia (CR)2, Universidad de Concepción, Concepción, CP 4030000, Chile
[4]Facultad de Ciencias del Mar y de Recursos Naturales, Universidad de Valparaíso, Valparaíso, CP 2362807, Chile
[5]Centro de Investigación Oceanográfica COPAS COASTAL, Universidad de Concepción, Concepción, CP 4030000, Chile
[6]Instituto Milenio en Socio-Ecología Costera, SECOS, Universidad de Concepción, Concepción, CP 4030000, Chile

*Correspondence to*: Reynier Bada-Diaz (rbada@udec.cl)

**Abstract.** Intraseasonal climate variability as, the Madden-Julian Oscillation (MJO), and synoptic-scale systems modify the normal conditions of the atmosphere and ocean, causing anomalies in sea surface temperature (SST) and salinity (S) which could create an environment conducive to algal bloom events in fjord systems, which in some cases can be toxic (HABs). In this work, an analysis of the atmospheric forcings on the synoptic-to-intraseasonal scale (SY-IS), that precede and proceeds to extreme high chlorophyll-a (chl-a) events was made in the Puyuhuapi fjord (44.7ºS 72.8ºW), during the summer season (December-February, DJF) between the years 2010-2018. Extreme events of high chl-a are defined when chl-a anomalies exceed the 90th percentile, and day 0 was defined as the maximum anomalous value. Six extreme events, corresponding to 83% of the total, were detected in the year 2016, a year with strong El Niño southern Oscillation (ENSO). From the analysis of the SY-IS patterns of persistent atmospheric anomalies during these 2016 events and their similarities, we detected that 4 events presented the characteristic of the passage of a low-pressure system, starting at least 7 days before the extreme chl-a event, with negative anomalies of sea level pressure and surface temperature, a change in wind direction and an increase in salinity at surface waters. we propose an atmospheric-oceanographic mechanism that induces favourable conditions for high phytoplanktonic activity in summertime: the passage of a low-pressure system, that weakens stratification and induces upwelling of deeper, colder and nutrient-rich waters favouring an increase in phytoplankton activity and the occurrence of extreme events of high chl-a in Puyuhuapi fjord. Furthermore, this work suggests that active phases 6 and 7 of the MJO might reinforce, on the SY-IS time scale, in DJF 2016. In the case of microalgae blooms, in addition to the well-known seasonal and interannual behaviors, it is important to superimpose the high-frequency variability. To improve the predictive ability of algal blooms and their relationship with climate conditions is essential for managing and mitigating their negative impacts on aquatic ecosystems, human health, and the economy.



32   **Graphical Abstract**



33



## 1 Introduction

Coastal waters of the southern fjord region in Chile experience changes in phytoplankton biomass and primary production due to climate and oceanographic variability (Iriarte et al., 2007; Pizarro et al., 2000; Saggiomo et al., 1994). This variability is controlled by the main limiting factors for marine productivity as mixing/stability, light and nutrients availability (nitrate and phosphate) (Jacob et al., 2014; Iriarte et al,. 2013). In the cases of Patagonia fjords, they exhibits strong seasonality: a productive season (September-November, SON) where the rate of nutrient supply could be the limiting factor and a non-productive season with light limitation (June-August, JJA) (Iriarte et al 2007; González et al., 2011; Montero et al., 2011; Iriarte et al., 2001). Therefore, seasonal variability can drive favourable conditions for phytoplankton biomass through gradual and persistent changes in sea level pressure (SLP), freshwater streamflow, temperature, incident solar radiation, and changes in wind speed and direction (Garreaud., 2018; Iriarte et al., 2017; Jacques-Coper et al., 2023).

Synoptic (days) and intraseasonal (25-80 days) variability (SY-IS) refers to changes in the ocean-atmosphere coupled system that manifest as oscillations in numerous variables such as intensity and direction wind, atmospheric pressure, sea surface temperature (SST), among others. From a climatic point of view, SY-IS variability represents high frequencies that break into the seasonal variability generating a wide range of responses on oceanographic and biological variables.

The Madden-Julian Oscillation (MJO) is the main mode of intraseasonal variability in the tropics and it is characterized by a large-scale convective dipole propagating from the eastward (Zhang, 2005). It has been found that the MJO can modify the extratropical circulation through teleconnections with high latitudes (Matthews & Meredith, 2004). Typically, the MJO exhibits periodicities between 30 and 90 days and has also strong seasonal variability, with a convective anomaly centred in the southern hemisphere during the austral summer (Peng et al., 2019). Large-scale impacts associated with this oscillation have been described for higher latitude climates, in South America. Jacques-Coper et al. (2015), analysed the relationship between the MJO and intraseasonal temperature in eastern Patagonia during DJF whereas Barrett et al. (2012) and Alvarez et al. (2016) described the influence of this oscillation on precipitation regimes and surface air temperature in South America, including Chile.

Regarding the impact of SY-IS on biological variables in the surface ocean, Gomez et al. (2017) highlight atmospheric perturbations associated with the MJO in Central Chile on phytoplankton biomass, showing positive (negative) Chl-a anomalies that coincide with patterns related to oscillation perturbations in phases 4-5 (8-1). Valdebenito et al. (2018) obtained a good correlation between heat waves and Alexandrium Catenella bloom in northern Patagonia for the period 1994-2014. Moreover, Narváez et al., (2019) described the influence of intraseasonal variations in biogeochemical conditions and water column mixing in northern Chilean Patagonia, as a result of a periodicity band of approximately 30 days, in this case, associated with the Baroclinic Annular Mode (BAM). Jacques-Coper et al. (2023) found that high biomass events in Inner Sea of Chiloé



occurred under the influence of a mid-latitude migratory anticyclone, inducing negative cloudiness (or increased
photosynthetically active radiation: PAR) and positive SST anomalies.

This study focuses on two timescales: seasonal and synoptic-to-intraseasonal (SY-IS). SY-IS variability, such as the Madden-
Julian Oscillation (MJO) to understand the influence of SY-IS variability on phytoplankton biomass, which is vital, considering
its potential modulation of extreme chl-a events ad HAB (Montes et al., 2018; León-Muñoz et al., 2018; Garreaud., 2018). In
particular, the present study will concentrate on extreme chl-a events in the Puyuhuapi Fjord area and explore how atmospheric
variables influence the hydrographic environment, favouring high phytoplankton biomass. The research aims to suggest a
mechanism by which SY-IS variability modulates water column conditions, triggering extreme chl-a events. We will assess
the influence of annual cycle, seasonal anomalies, and SY-IS anomalies (particularly the MJO) on modifications in the
hydrographic environment. This knowledge can aid in understanding extreme events and mitigating their impacts. Section 2
will introduce the study area, data, and methodology; section 3 will present results, including climatology, extreme chl-a events
in DJF, and the possible link of MJO modulation with extreme events. Finally, section 4 will discuss and summarize the
investigation's findings.

**2 Data and methods**
**2.1 Study area**
The Puyuhuapi Fjord (PF) is located around 44.7ºS 72.8ºW, in the Aysén region, in northern Chilean Patagonia (Fig. 1). With
an area of approximately 700 km$^2$, it connects with the Jacaf Fjord in the north, which extends towards the Moraleda Channel,
which in turn connects with the Pacific Ocean. To the south, the PF connects directly to the Moraleda Channel. This fjord
receives freshwater inputs from different rivers that flow into it, mainly from the Cisnes River (44.74ºS 72.71ºW), which has
an average annual flow of 218 m$^3$/s (Calvete & Sobarzo, 2011). In the winter months, the highest chl-a values are observed in
the fjord, because nutrient availability is higher with respect to summer conditions due to the fact that the water column is
partially mixed and there is a weakening of stratification (Schneider et al., 2014).



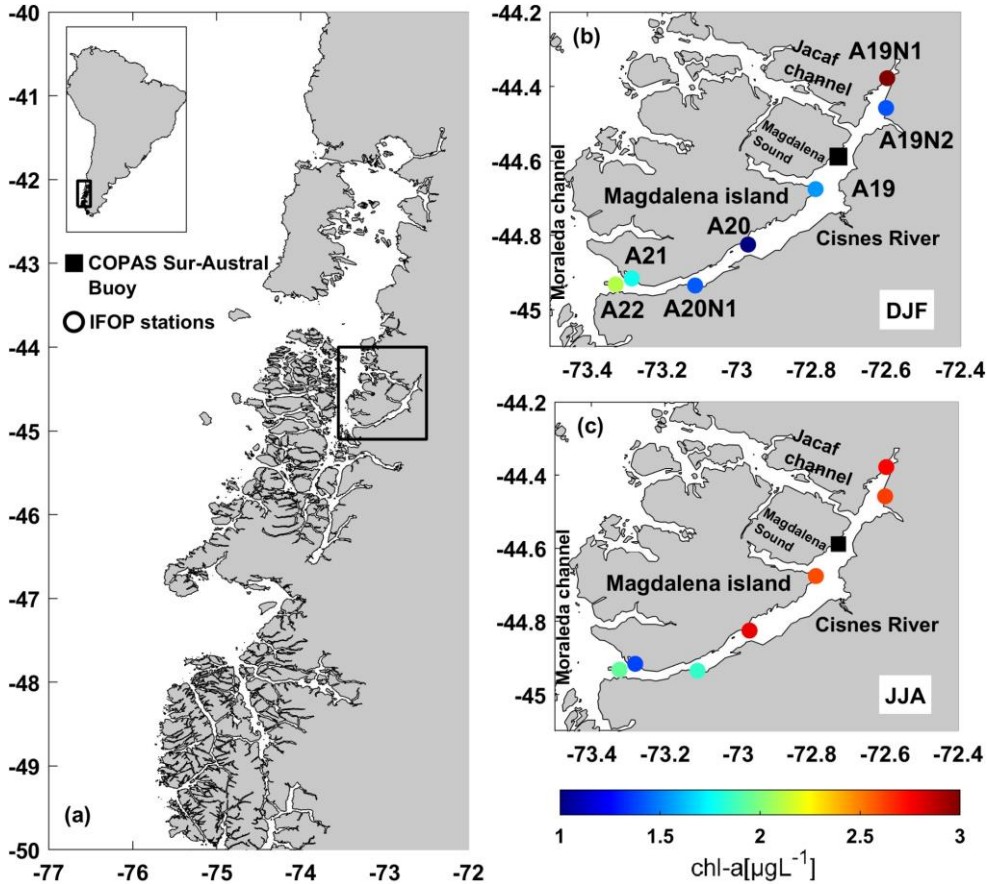

**Figure 1: Map of the study area. a) Northwest Patagonia and its location in South America (small top box), b) area surrounding the Puyuhuapi fjord; the colored dots indicate the summer (DJF) chl-a averages measured by sampling stations (see text) for the period DJF 2012-2020, and c) same as panel b, but for winter (JJA). The black square indicates the location of the buoy (see text).**

The intrusion of oceanic waters into the fjord is mainly dominated by the seasonal variability of the large-scale wind field, in turn induced by the latitudinal migration of the South Pacific High (SPH) (Schneider et al., 2017). In winter, the core of the SPH reaches the lowest latitudes of its annual cycle, clearing the way for westerly winds over the study area. These favor the advection of oceanic waters into the PF through the two connections with the open sea and there is greater water input from the ocean into the fjord (Schneider et al., 2014). In summer, the high pressure over the study area (Fig. 2, panel a) is due to the southward shift of the SPH core, a predominant feature in the spring and summer seasons, when it is more intense at approximately 35° S and extends to 45° S (Ancapichún & Garcés-Vargas, 2015). In these seasons, the predominance of high pressure and anticyclonic circulation over the area prevent the arrival of frontal systems and reduce precipitation and cloudiness (Sobarzo et al., 2007). Also, the intrusion of oceanic waters and nutrients may be dominated by wind variability on the synoptic scale, due to the effects of passing low and high pressure systems (Pérez-Santos et al., 2019).



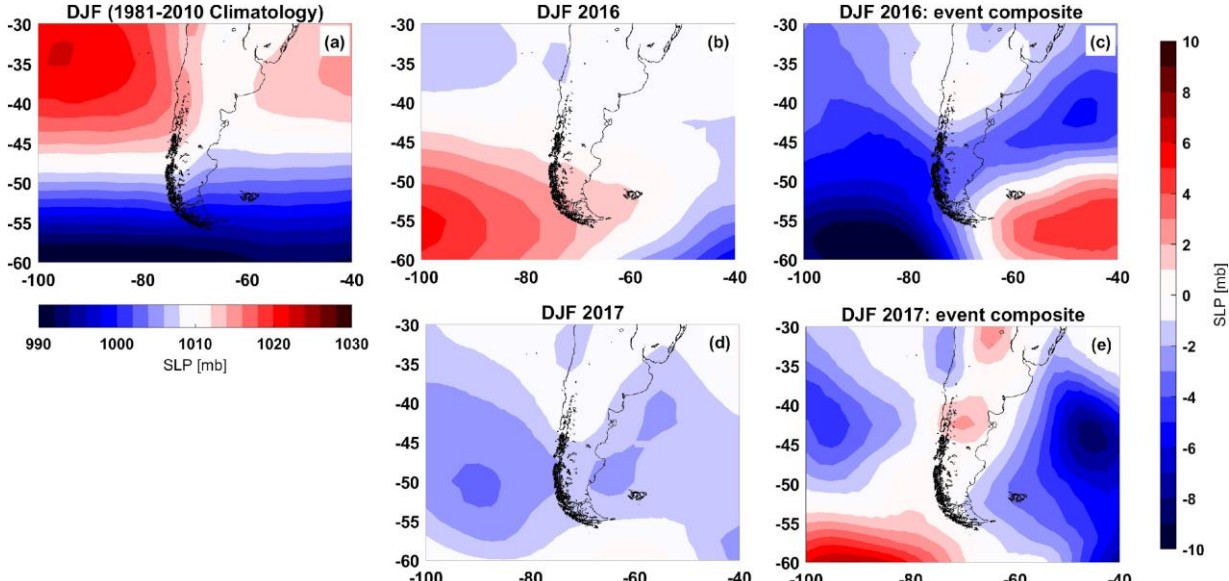

**Figure 2: Sea level Pressure (SLP) fields: a) 1981-2010 summer climatology, b) seasonal anomaly of DJF 2016 with respect to climatology, c) composite of daily anomalies for DJF 2016 considering (28-Dec-2015, 28-Jan-2016, 19-Feb-2016 and 27-Feb-2016), d) seasonal anomaly of DJF 2017 with respect to climatology e) composite of daily anomalies for DJF (7-dic, 19-dic y 28-dic de 2016). Source: NCEP/NCAR reanalysis.**

The characteristics of the water column in the fjord themselves also show seasonal variability, with a strong haline and thermal stratification observed mainly in summer, due to the influence of freshwater discharged by rivers throughout the fjord area and the relatively higher water temperatures due to the increase in solar radiation predominant during this season (Schneider et al., 2014). This is corroborated in section 3.1 (climatology of study area).

In this study, the level of vertical stratification of the water column was calculated using $n_s$ parameter proposed by Haralambidou et al., 2010 as follows:

$$n_s = \frac{S_{20m} - S_{1m}}{0.5(S_{20m} + S_{1m})} \tag{1}$$

Where S corresponds to the salinity values at the surface ($S_{1m}$) and at 20 meters ($S_{20m}$). For $n_s < 0.1$, the water column is fully mixed, for $0.1 < n_s < 1$ it is partially mixed, and $for\ n_s > 1$ it is highly stratified (Haralambidou et al., 2010).

**2.2 In situ observations**

The extreme chl-a concentration events in this study were identified from observations corresponding to a YSI EMM2000 buoy located at 44º35'17" S - 72º43'37" W (Bpuy), deployed by the Centro de Investigación Oceanográfica COPAS Sur-Austral (Fig. 1a, Table 1). The observations correspond to hourly data of temperature (T), salinity (S) and chl-a, recorded by a multiparameter YSI 6000-V4 sonde.



To illustrate the chl-a, T and S seasonal variability throughout the fjord, CTD data for the period 2012-2020 were used (Fig. 1. Table 1). The data was collected as part of the Undersecretary of Fisheries and Aquaculture (Subsecretaría de Pesca y Acuicultura, SUBPESCA) Red Tide Management and Monitoring Program in the regions of Los Lagos, Aysén and Magallanes in agreement with the Instituto de Fomento Pesquero (IFOP). These measurements cover between surface and 50m depths, with variable frequency of data collection. The CTD data was seasonal averaged for each monitoring station, where JJA and DJF values correspond to winter and summer seasons.

**Table 1: In situ observations used in this study.**

| Source | Location | Station name | Latitude | Longitude | Period | Number of days with values (DJF-JJA) |
|---|---|---|---|---|---|---|
| COPAS Sur-Austral | Puyuhuapi Fjord (PF) | Puyuhuapi buoy | -44°35'17" | -72°43'38" | 2010-2018 | Daily |
| Subpesca-IFOP | San Andrés Island - PF | A22 | -44°55'57" | -73°19'28" | 2012-2020 | 23-19 |
| Subpesca-IFOP | Amparo Port – PF | A21 | -44°55'00" | -73°16'54" | 2012-2020 | 23-12 |
| Subpesca-IFOP | Uspallante – PF | A20N1 | -44°56'06'' | -73°06'41'' | 2012-2020 | 23-17 |
| Subpesca-IFOP | Marta Lighthouse – PF | A20 | -44°49'30" | -72°58'10" | 2012-2020 | 23-18 |
| Subpesca-IFOP | Magdalena Sound – PF | A19 | -44°40'34" | -72°47'17" | 2012-2020 | 20-16 |
| Subpesca-IFOP | Ventisquero Sound - PF | A19N2 | -44°27'29'' | -72° 35'57'' | 2012-2020 | 23-14 |
| Subpesca-IFOP | Ventisquero Sound - PF | A19N1 | -44°22'43'' | -72°35'46'' | 2012-2020 | 23-18 |

## 2.3 ERA5 Reanalysis and other data

To spatially characterize the environmental variability around the study area, fields with a horizontal resolution of 0.25° × 0.25° of SST, air temperature (t2m), zonal (u) and meridional (v) wind at 10 m , incoming radiation (mean surface downward short-wave radiation flux, Rad) and SLP, were retrieved from the European Center for Medium-Range Weather Forecasts



(ECMWF) ERA5 (Compo et al., 2011) reanalysis for the period 1981-2010. ERA5 is a state-of-the-art product, which
combines instrumental records with computational modeling to reconstruct fields of atmospheric variables in time and space.
To evaluate the possible influence by the tides on the occurrence of extreme events of high chl-a, we use sea level prediction
data for the southern area of the Puyuhuapi fjord obtained from the CDOM (Oceanographic and Meteorological Data Center
http://www.cdom.cl/index.php) made available by the COPAS Sur-Austral Oceanographic Research Center (COPAS Sur-
Austral).
The streamflow values of the Cisnes River correspond to a hydrological station located at 44º45'0" S - 72º43'0.12" W, which
contains daily streamflow values. These data were obtained from CAMELS-CL (Catchment Attributes and Meteorology for
Large Sample Studies, Chile Dataset) (https://camels.cr2.cl/), managed by Center for Climate and Resilience (CR2) (Alvarez-
Garreton et al., 2018).

**2.4 Intraseasonal anomalies**
To explore the SY-IS variability, SY-IS anomalies were calculated for each variable using Eq. (2) proposed by Cerne and Vera

152 (2011):

$$intraseasonal\ anomaly_{d,y} = daily\ values_{d,y} - climatological\ daily\ mean_d \qquad (2)$$

$$- \left( DJF\ seasonal\ mean_y - long-term\ DJF\ seasonal\ mean \right)$$

As can be seen, these anomalies are the result of subtracting from the daily values their corresponding daily climatological
means (thus eliminating the annual cycle) and the difference between the seasonal mean for that particular year and the long-
term seasonal mean (eliminating the year-to-year variability). This calculation was made separately for two seasons of the
year, that is, DJF and JJA. The averages were calculated from the full extent of the data, i.e. 2010-2018. From the SY-IS
anomalies, extreme chlorophyll events were defined according to criteria of intensity and persistence. Regarding intensity, the
90th percentile threshold (p90) was defined, calculated for each season of the year considering the whole period. In this way,
10% of the days were selected, corresponding to the largest outliers (chl-a anomaly value $> 9.1\ \mu gL^{-1}$). Considering the selected
days, a persistence criterion was applied to define the beginning of an extreme chlorophyll event, specifically the first day of
exceedance of the p90 threshold for at least 2 consecutive days; the end of the event, is reached when the anomalies remain
below p90 for at least 2 consecutive days. The day of the maximum value of chlorophyll anomalies during an extreme event
was designated as "day 0".
The sequence of synoptic settings prior to the extreme events of chl-a was analyzed using the SY-IS anomalies of the reanalysis
fields. For each event of interest, synoptic maps were made for the area -42º to -45.5º S and -72º and -75º W, covering 10 days
before day 0. Also, considering those events that exhibit similar synoptic conditions, a sequence of composites (average fields)
was calculated.



## 2.5 Madden–Julian Oscillation (MJO) index

To explore a possible association between local conditions and tropical activity characterized by the MJO, the real-time multivariate MJO index (RMM, Wheeler & Hendon, 2004) is used. This index, available from 1974 on, was obtained for the study period from the website (http://www.bom.gov.au/climate/mjo) of the Bureau of Meteorology of Australia. The RMM index is made up of two main components EOFs (RMM1 and RMM2) extracted from the zonal wind at 850 and 200 hPa and interpolated satellite long-wave radiation. Therefore, it can be plotted in a two-dimensional phase space defined by these orthogonal axes and characterized by amplitude and phase values, which describe the intensity and position of the convection centers in the tropical band, respectively. Wheeler et al., 2004, divided this phase space into eight equal segments and stated that the nominal transition time between each of them is 6 days, but with variations from event to event. The active phases of the oscillation correspond to those days in which the index exhibits an amplitude greater than 1. With the data of this index, a phase diagram was made with the trajectories of the MJO for each trajectory of days leading to the identified extreme chl-a events. Such trajectories depict the behavior of the oscillation 20 days before each of the extreme events. Finally, with the aim of characterizing the potential modulation by the MJO of the physical environment of the study area, the ERA5 variables were used to characterize a climatology of the intraseasonal anomalies, considering composites for each of the active phases of the MJO for the study period 1981-2010.

## 3. Results

### 3.1 Climatology of Puyuhuapi fjord

The seasonal long-term average for surface chl-a shows the highest values ~3 µgL$^{-1}$, in summer in station A19N1, which is located at the northeastern most position of the fjord (Fig. 1b). Similarly, higher JJA chl-a average values are found at stations closer to the Jacaf Fjord (A19N1 and A19N2, Fig. 1c).





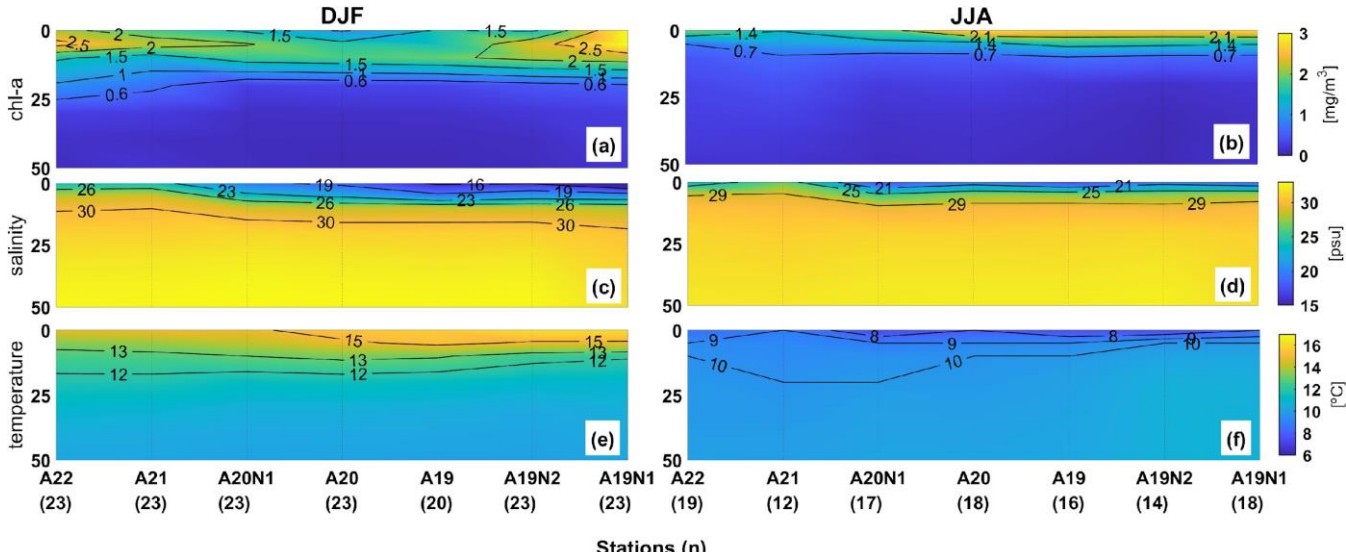

**Figure 3: Mean DJF and JJA values of in-situ chl-a, salinity (S) and temperature (T) at different stations across the Puyuhuapi fjord. Number of days with values: n. Source: see text and Table 1.**

Figure 3 shows the DJF and JJA mean conditions of the water column along the fjord, based on chl-a, T and S observations from the Subpesca-IFOP program (Table 1). Again, DJF shows high chl-a values in the northeasternmost stations of the fjord (sampling stations A19N1 and A19N2), with values reaching ~3 μg L$^{-1}$ (Fig. 3a) and extending down to 10 m depth from the surface. Station A22, which is located closer to the Moraleda Channel, shows on average high chl-a values, exceeding 2 μgL$^{-1}$ at 5 m. The climatological hydrography shows difference between winter and summer, in winter temperature shows values of 8ºC at the surface and 5º C from 15 to 50 m and salinity at the surface has values of 21 psu and 29 psu at 5 m. In summer the highest T values are above 16° C at the surface, and S shows the lowest values in the first 10 m, with values of 16 psu at the surface (Fig. 3c). In JJA, a weakening of the stratification due to a lower proportion of low S waters and partially-mixed water, and T are lower and the marked stratification is not seen.

The time series of T, S and chl-a from mid-2010 to February 2018 are shown in each of the panels of Figure 4. The seasonal cycle of T is well marked (Fig. 4a), reaching values of up to ~20º in DJF, when radiation is maximum. S (Fig.e 4b) shows the highest values in MAM-JJA. The highest S values are recorded in late SON and early JJA 2016, with positive anomalies reaching ~30 psu and a maximum value of 31.8 psu on May 27 (Fig. 4b). The highest chl-a values are observed, on average, mainly in the MAM-JJA seasons (Fig. 4c), when high S values and low T values are also apparent. However, throughout 2016, high values are observed during the DJF and MAM months.





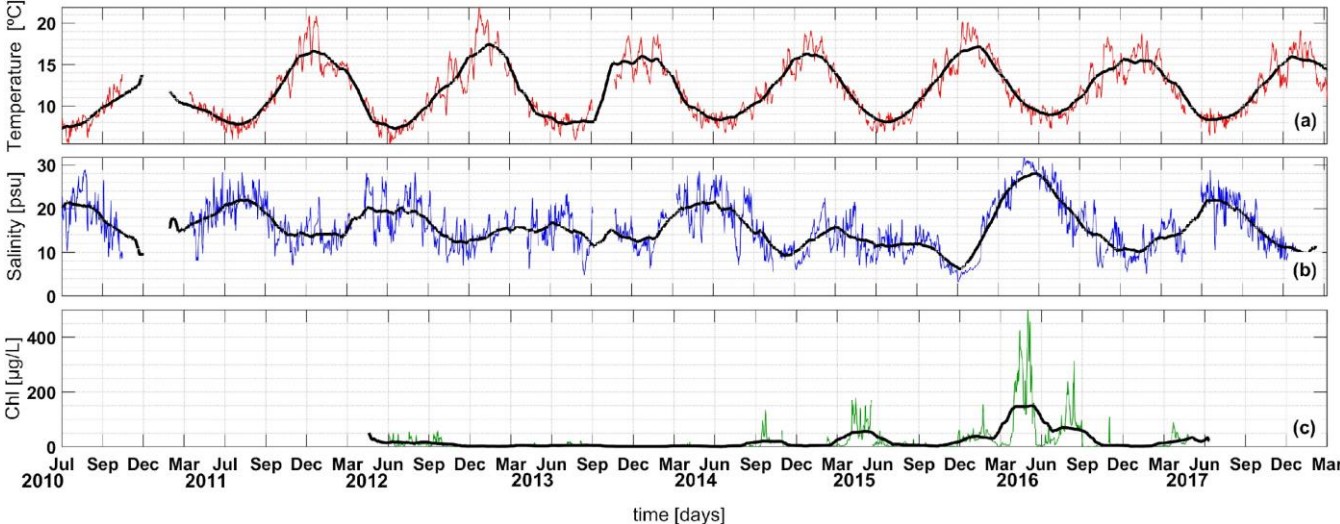

**Figure 4: Daily observations of a) surface temperature (T), b) salinity (S) and c) Chlorophyll-a (chl-a) from the buoy located in the Puyuhuapi fjord in the period 2010-2018. Color lines indicate daily averages, black lines indicate the signal smoothed by the loess method (1 cycle per 30-time increments).**

Close to the Puyuhuapi buoy in the sampling station A19, highest values of the $n_s$ parameter (equation 1), ~0.8 units, in SON-DJF were detected (Table 2). A decrease in $n_s$ is observed at the stations located further southwest, mainly at station A21, with values of 0.3 and 0.17 at DJF and JJA, respectively.

**Table 2: Haralambidou parameter measuring vertical stratification for each sampling station in summer (DJF) and winter (JJA).**

| Sampling station | DJF | JJA |
| --- | --- | --- |
| A19N1 | 0,96 | 0,60 |
| A19N2 | 0,73 | 0,45 |
| A19 | 0,75 | 0,62 |
| A20 | 0,60 | 0,51 |
| A20N1 | 0,56 | 0,69 |
| A21 | 0,30 | 0,17 |
| A22 | 0,32 | 0,33 |

**3.2 Extreme events of high chl-a in summer (DJF)**

The distribution of daily values of chl-a in the period of 2010-2018 of Bpuy corresponding to the study it is log-normal, with an average of 8 µg L$^{-1}$ and a standard deviation of 17.6 µg L$^{-1}$ over the entire period. The p90 threshold corresponds to 9.1 µgL$^{-1}$, i.e., somewhat more than three times the average summer value. A total of 6 extreme events were found that met the





intensity and persistence criterion (see section 2.4). Figure 5a-d shows the evolution, centered on day 0, of chl-a, temperature,
salinity and streamflow of Cisnes River from 10 days before to 10 days after each event (thin black lines), together with the
respective mean signals (thick black line). Chl-a anomalies (Fig. 5a), on average, show values below p90 on day -10, remaining
so until day -5, where they exceed the threshold and increase until day 0, where the anomalies reach their highest value (46.9
$\mu$gL$^{-1}$); then, they decay until day 4, reaching negative anomalies on day 10 (-5.0 $\mu$gL$^{-1}$). As shown in Figure 5b-c, on day -10,
T and salinity show anomalies of 0.2ºC and -2.4 psu respectively; on day 1 and -8, they reach negative anomalies of -0.5ºC
and -3.3 psu, respectively; towards day 0, the trend of T (salinity) is to decrease (increase). On day -8, a strong tendency to
decrease in T (Fig. 5b), and to a lesser extent also to increase in salinity. Figure 5d shows streamflow anomalies, a decrease
can be observed between days -10 and -7, with anomalies reaching -50 m$^3$/s, from day -7 an average increasing trend is denoted
until day 0, with anomalies exceeding 65 m$^3$/s. This shows an increasing trend between days -10 and 0. As can be seen in Table
3, the event corresponding to February 19, 2016 reaches the highest record of chl-a anomaly (100.1 $\mu$gL$^{-1}$), with high negative
(positive) anomalies of temperature (salinity) that persist until day +10.

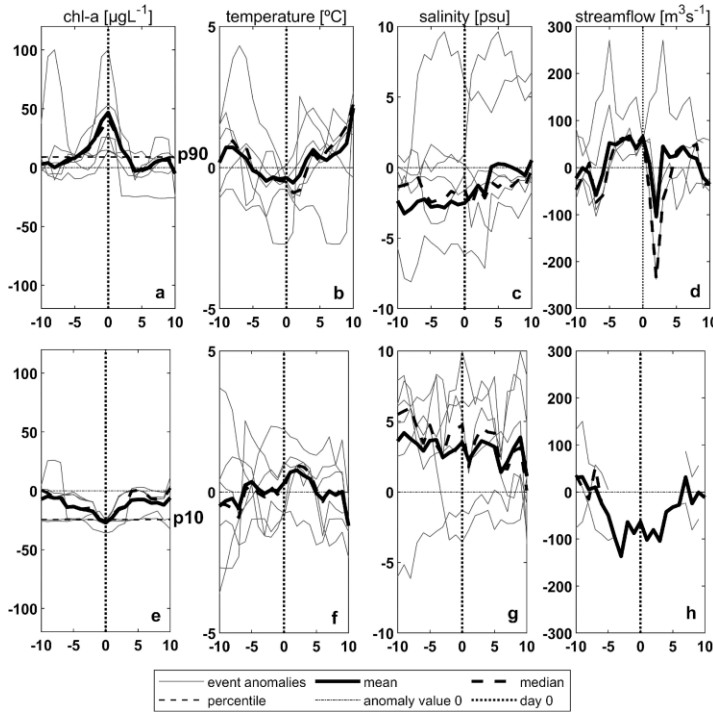


**Figure 5: Intraseasonal anomalies of a) chl-a, b) temperature (T), c) salinity (S), and d) Cisnes River streamflow in extreme events of high (upper row, a-d) and low (lower row, e-h) chl-a values in DJF for the Puyuhuapi buoy. Day 0 indicates the highest (lowest) anomalous value of the event. The gray lines show the different events, the thick and dotted black lines indicate the mean and median respectively for the different variables.**

**Table 3: Extreme events of high chl-a during DJF recorded at the Puyuhuapi buoy (2010-2018). The date (day 0), the daily values of chl-a, their corresponding SY-IS anomalies, and the MJO phase and intensity on day 0 are shown.**




| Season | Date | Daily chl-a value ($\mu gL^{-1}$) | Chl-a anomalies ($\mu gL^{-1}$) | MJO phase | Index WH-MJO |
|---|---|---|---|---|---|
| DJF | 12/07/2014 | 21.9 | 14.9 | 6 | 1.9 |
| | 12/28/2015* | 62.9 | 53.3 | 6 | 2.6 |
| | 01/16/2016 | 35.9 | 14.2 | 3 | 1.2 |
| | 01/28/2016* | 50.7 | 29.8 | 4 | 1.2 |
| | 02/19/2016* | 149.3 | 100.1 | 7 | 1.6 |
| | 02/27/2016* | 63 | 26 | 7 | 2.1 |


A summary with the main characteristics of these events is presented in Table 3, with the date, the absolute values, the chl-a
SY-IS anomalies for the days of occurrence of the extreme events in DJF, the phase in which the MJO was found, and the
intensity of the WH-MJO index. Of the total of 6 extreme chl-a events found in the summer of the time series, 5 (i.e., ~83%)
correspond to the summer of 2016. An exhaustive analysis of the 6 events in Table 3 was performed; subsequently, 4 of these
events were examined in detail. They presented similar synoptic sequences in the period of 10 days before their occurrence,
with negative SLP anomalies predominating over PF.

**3.2.1 Summer 2016**

Figure 2b shows the SLP anomalies for DJF 2016 relative to the 1981-2010 climatology. Positive anomalies centered around
55ºS and extending to the entire southern tip of Patagonia, including Tierra del Fuego, are observed. This configuration
corresponds to an atmospheric blocking due to an anticyclone that predominates from spring 2015 to autumn 2016. Very
different conditions were found on the days of occurrence of the 4 similar extreme events with respect to anomalies on DJF
2016 (events marked with * in Table 3) (Fig. 2c).
The correlation of the anomalies of the analyzed variables during DJF 2016 (values shown in table S1 of the supplementary
material), show a direct relationship between SLP and T (R~0.26, $p < 0.05$). The SLP and chl-a anomalies show also an inverse
relationship, similar to the previous one (R~-0.24, $p < 0.05$). The first event of DJF 2016, which occurred on December 28,
2015 (day 0), is characterized by reaching anomalies of 53.3 $\mu gL^{-1}$, -0.5ºC, and -6.2 psu on day 0 (Fig. 6b-c). The event that
reached the highest chl-a value of 100 $\mu gL^{-1}$ during this summer occurred on February 19 (day 0) (Fig. 6d). During this event,
negative T anomalies and positive salinity anomalies of approximately 1 psu prevailed from day -9. This event starts
approximately on day -8 by exceeding the threshold of p90; thereafter, it remains 11 days (February 11 to February 21) above
it. During the previous days (day -9 to -3), a decreasing trend in T is observed, with anomalies that start at 1.3ºC and reach -
1.7ºC (Fig. 6b) and an increase in salinity anomalies from -0.5 psu on day -8 to 0.4 psu on day -1 (Fig. 6c). The increase in



chl-a occurs between days -5 and 0, when the highest anomaly (100 µgL$^{-1}$) is reached. In this period, temperature and salinity
anomalies range from 0.1 to -0.7ºC and -0.3 to 0.9 psu respectively.
In Figure. 6, dotted lines indicate the relevant trends of the variables observed, calculated 10 days prior to the occurrence of
extreme events of DJF 2016. In general terms, a decreasing trend of SLP anomalies before the occurrence of four of the five
extreme events of DJF 2016 is evident (*Table 3, Fig. 6a). This signal corresponds to eastward-moving cold fronts over the
study area. Consistently, the T also shows a decreasing trend (Fig. 6b), which is more intense in the event around January 16,
2016. Salinity (Fig. 6c) shows an increasing trend before the occurrence of 4 out of 6 extreme events in DJF 2016. This
phenomenon is more evident in the last event of DJF 2016, corresponding to the day February 27.

As indicated in Figure 6e, all 5 events found in DJF 2016 occur during active phases of the MJO, as the cycle progresses from
phase 6 (starting on December 28, 2015) to phase 7 (starting on February 27, 2016). The event with the largest anomalies of
chl-a occurs during phase 7 (extending from February 15 to 18), just after the transition from the active phase 6, in which it
was the previous 4 days (Fig. 6e).

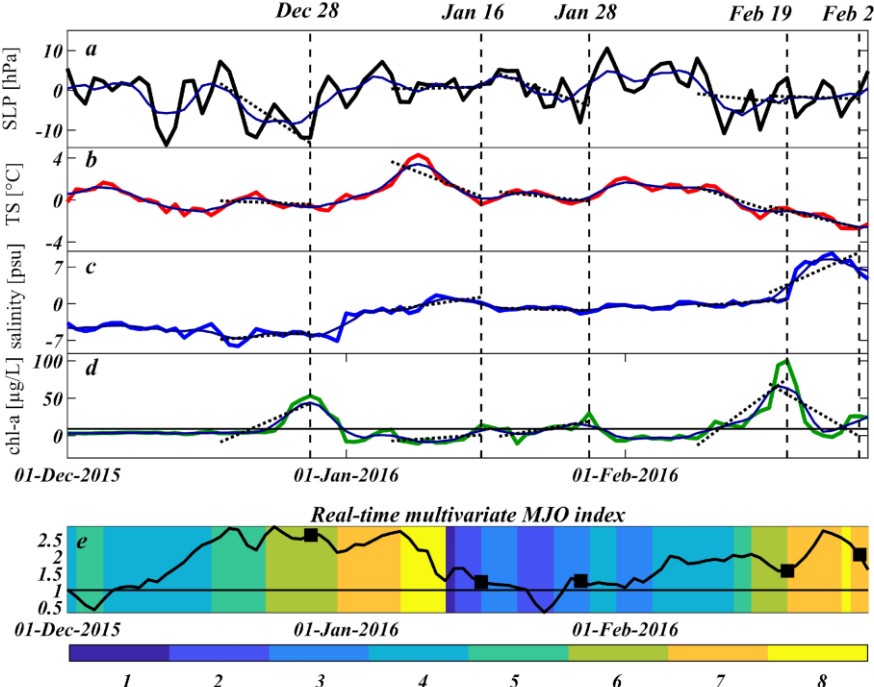


**Figure 6: Daily SY-IS anomalies of a) sea level pressure (SLP), b) temperature (T), c) salinity (S) and d) chl for DJF 2016. The blue line shows the values smoothed by a 5-day moving average, the vertical black lines indicate the days of occurrence of the 5 extreme events corresponding to that summer, and the dashed black lines indicate the calculated trend 10 days before the occurrence of the events. e) shows the amplitude (black curve) and the phase (background color) of the MJO index; black squares indicate the days of occurrence of the events.**



As evident from Figure 6, 4 of the 5 extreme events found in DJF 2016 exhibit similar trends in the behavior of SLP, T and S.
This similarity consists of a decreasing trend of SLP and T in the 10 days preceding day 0, and increasing S. For this reason,
the synoptic evolution of these events showing similar tendency trends (28-Dec-2015, 28-Jan, 19-Feb and 27-Feb-2016) is
explored below. The 28-Dec-2015 event is selected as representative of this subset, and its corresponding sequence of fields
is shown in the SM.
The sequence of composites of the 4 selected events (28-Dec-2015, 28-Jan, 19-Feb, and 27-Feb-2016) shown in Figure 7
exhibits the changes in the previous 10 days of the event, of the SY-IS anomalies of SLP (black contours), t2m (shading),
radiation (contour) and wind direction and magnitude (vectors) of ERA5 over the study area. From day -10 to -8, positive SLP
and SW wind anomalies, reinforcing the prevailing seasonal conditions. On day -7, we observe the presence of negative SLP
anomalies that persist until day 0 and N wind anomalies, indicating the approach of a low-pressure system a week before day
0 (the figures of individual selected events show in the supplemental material, S1-4). Incoming radiation anomalies indicate
the passage of this frontal system, with negative anomalies from day -8 to day 0, due to increased cloudiness, while days -10
and -9 show positive anomalies, which coincide with the positive SLP anomalies. T2m shows no changes between days -9 and
-7, while between days -6 and 0 negative anomalies predominate with values reaching -2°C on average for the PF area (Fig.
7). On average, for the 10 days, a predominance of negative anomalies of incoming radiation and t2m is observed, the latter
with anomalies around -3ºC throughout the PF zone (Fig. 7). The sequence of composites for the 4 events shows the passage
of a low-pressure system over the area surrounding the Puyuhuapi fjord, showing a predominance of negative values of SLP,
t2m and incoming radiation (Fig. 6-7).



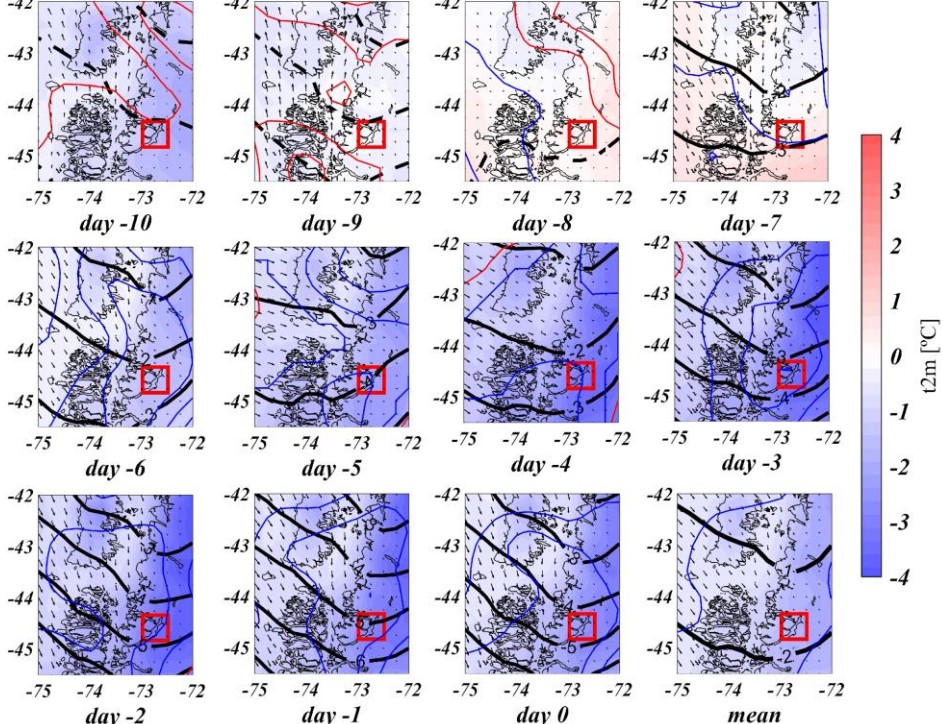

**Figure 7: Sequence of daily composites of intraseasonal anomalies, starting 10 days before the occurrence of the extreme events of high chl-a that peaked on 28-Dec-2015, 28-Jan, 19-Feb, and 27-Feb-2016. Each panel corresponds to one day and the last panel corresponds to the average of the 10-day composites of anomalies. Colors indicate t2m, blacks contours indicate SLP every 1 hPa (continuous contours: negative anomaly, dashed contours: positive anomaly), color contours indicate incoming solar radiation every 20 W/m2 (red: positive anomaly, blue: negative anomaly) and vectors indicate wind speed. Source: ERA5 reanalysis.**

## 3.3 Extreme events of low chl-a in summer (DJF)

To explore the potential linearity of the relationship between atmospheric conditions and the concentration of chl-a in Bpuy, we contrast the results obtained regarding extreme events of high intraseasonal anomalies of chl-a (subsection 3.2) with the conditions prevailing during opposite events. Specifically, we performed an analysis like the previous one, but focused on extreme events of low chl-a anomalies identified within the same period (2010 - 2018). In a similar fashion as before, events are defined as those lying below the intensity threshold corresponding to the 10th percentile (p10= - 24 µgL⁻¹) and exhibiting the same persistence condition used above. In this way, 6 extreme events were found (Table 4). Figure 5e-h shows that the mean signal of chl-a anomalies are above p10 from day -10 to day -7. Thereafter, a strong drop is observed with negative anomalies of -2 µgL⁻¹ on day -7 to -22 µgL⁻¹ on day -4; then, decaying below p10, until day 0, reaching negative anomaly values of -27.2 µgL⁻¹ (Fig. 5e). T starts from a negative anomaly (-0.5°C) on day -10 (Fig. 5f) and shows an increasing trend until day 2, when it reaches a positive anomaly of 0.8°C; later, it declines until day 10 (-1.2°C). As shown in Figure 5g, salinity exhibits a decreasing trend, with a positive anomaly of 3.6 psu on day -10 and reaching the lowest mean anomaly value on day 10 (0.1 psu). Streamflow anomalies show a decreasing trend from day -10 (32 m³/s) to -3 (-139 m³/s); then, increasing until





day 10 (Fig. 5h), when anomalies are close to 0. Nevertheless, this trend in the streamflow anomalies is not reliable due to the
absence of data on the dates corresponding to extreme events of low chl-a.

**Table 4: Extreme events of low chl-a during DJF recorded at the Puyuhuapi buoy (2010-2018). The date (day 0), the daily values of**
**chl-a, the values of the SY-IS anomalies, and the corresponding MJO phase and intensity are shown.**

| Season | date | Daily chl-a value (µgL$^{-1}$) | Chl-a anomalies (µgL$^{-1}$) | MJO phase | Index WH-MJO | |
|---|---|---|---|---|---|---|
| | 2/19/2013 | 1.3 | -26.1 | 4 | **1.7** | |
| | 2/19/2014 | 1.2 | -25.4 | 6 | **1.9** | |
| DJF | 2/19/2015 | 3.6 | -24.6 | 7 | **0.3** | |
| | 12/07/2016 | 0.8 | -26.2 | 2 | **0.6** | |
| | 12/16/2016 | 1.4 | -24.9 | 6 | **0.6** | |
| | 12/28/2016 | 2 | -36 | 5 | **0.7** | |


### 3.3.1 Summer 2017

Of a total of 6 events with negative anomalies, 3 correspond to DJF 2017. In this season, a predominance of negative SLP
anomalies is observed in the southeastern Pacific off Chilean Patagonia, around 40ºS and 95ºW (Fig. 2d), with anomalies
reaching -3 mb surrounding the Puyuhuapi fjord. Such conditions are very different from those of the previous summer (DJF
2016), during which positive SLP anomalies predominated (Fig. 2b). During the DJF 2017 events, SLP anomalies over the
Puyuhuapi fjord are neutral. Centered on the eastern Inner Sea of Chiloé (∼ 42ºS-45ºS) a positive anomaly of 2 mb are
observed; also, around 60ºS -90ºW an anticyclone can be seen (Fig. 2e).
The event with the lowest recorded negative chl-a anomaly of -36 µgL$^{-1}$, occurred on 28-Dec-2016 (Fig. 8). The environmental
conditions prior to the event exhibit positive SLP anomalies on days -10 and -1, with values of 1.9 and 3 hPa, respectively.
From day -7 on, a positive trend is observed in T (Fig. 8a-b). Additionally, from day -5 a negative trend in S is observed (Fig.
8c). In general, the trends in SLP, T and S are opposite to those shown for extreme events of high chl-a (Fig. 6). The MJO was
mostly inactive during December 2016 (Fig. 8c).

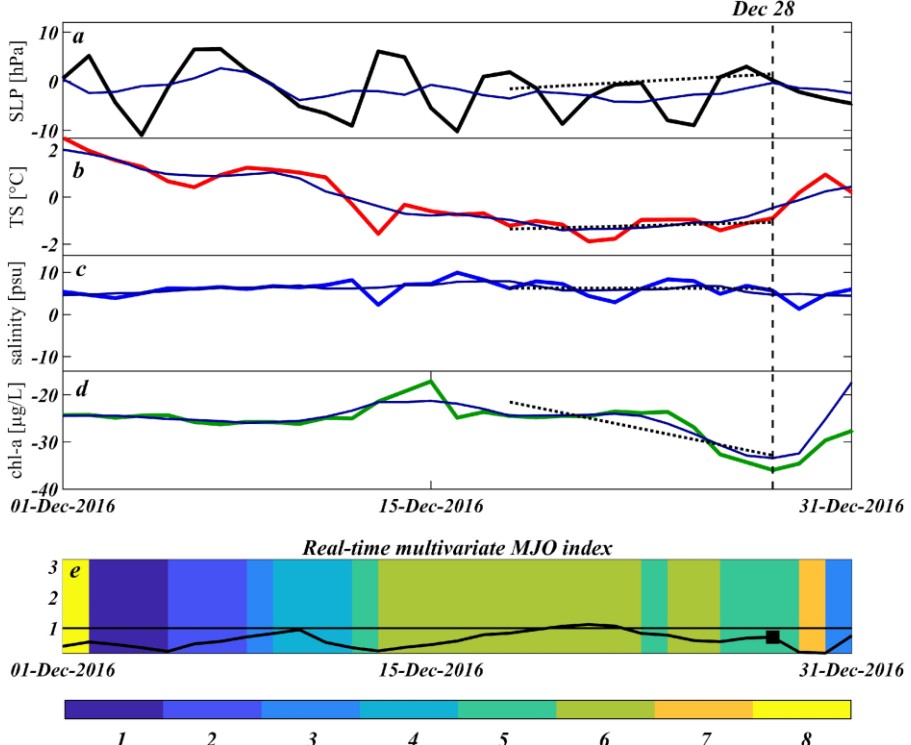

**Figure 8: Daily intraseasonal anomalies of a) sea level pressure (SLP), b) temperature (T), c) salinity (S) and d) chl-a for December 2016. The blue line shows the values smoothed by a 5-day moving average, the vertical black lines indicate the days of occurrence of 28-Dec-2016 extreme event, and the dashed black lines indicate the corresponding trends starting 10 days before the occurrence of the events. e) shows the amplitude and phases of the MJO index. Black squares indicate the days of occurrence of the events.**

The analysis of the composite synoptic sequence prior to the event peaking on 28-December-2016 (Fig. S5 in SM) shows a predominance of positive SLP anomalies from day -2 to day 0, N wind anomalies on most days, and negative incoming radiation and t2m anomalies during most of the sequence. In this composite we did not find any characteristic pattern associated with any synoptic phenomena.

## 3.4 Madden–Julian Oscillation (MJO)

To analyze the possible modulation that the MJO may exert on the atmosphere-ocean interaction in the PF, phase diagrams were constructed to show the MJO trajectories preceding each of the 6 extreme events of high chl-a in Puyuhuapi fjord (described in section 3.2, Table 3), considering the respective periods between days -20 and 0 (Fig. 9a-b). It is noteworthy that all 6 extreme events peak on day 0 during active phases of the MJO (Table 3). Two subgroups are observed, the first one peaking on day 0 during active phases 6 and 7 (4 events), and the second one peaking on phases 3 and 4 (2 events). The individual trajectories show differences, the event peaking on 07-Dec-2014 during an active phase 6 was preceded by non-active MJO conditions on day -20, then it goes through active phases 3, 4 and 5, until day 0. The events corresponding to 28-Dec-2015, 19-Feb-2016 and 27-Feb-2016 show similar trajectories, starting on day -20 during an active phase 4, and evolving




towards active phases 6 or 7 (Fig. 9a). The events of 19-Feb and 27-Feb-2016 overlap with a span of 8 days, as detailed in
Figure 10a. As shown in Table 3, the event peaking on 16-Jan-2016 occurs during an active phase 3 of the MJO, but 20 days
earlier, the MJO showed active phase 6 conditions, evolving through active phases 7, 8, 1, and 2 in between (Fig. 9b). The
event peaking on 28-Jan-2016 shows a trajectory that starts in an active phase 8 on day -20, evolving through active phases 1
and 2; then, it evolves towards a non-active phase 3, until reaching active phase 4 on day 0 (Fig. 9b).

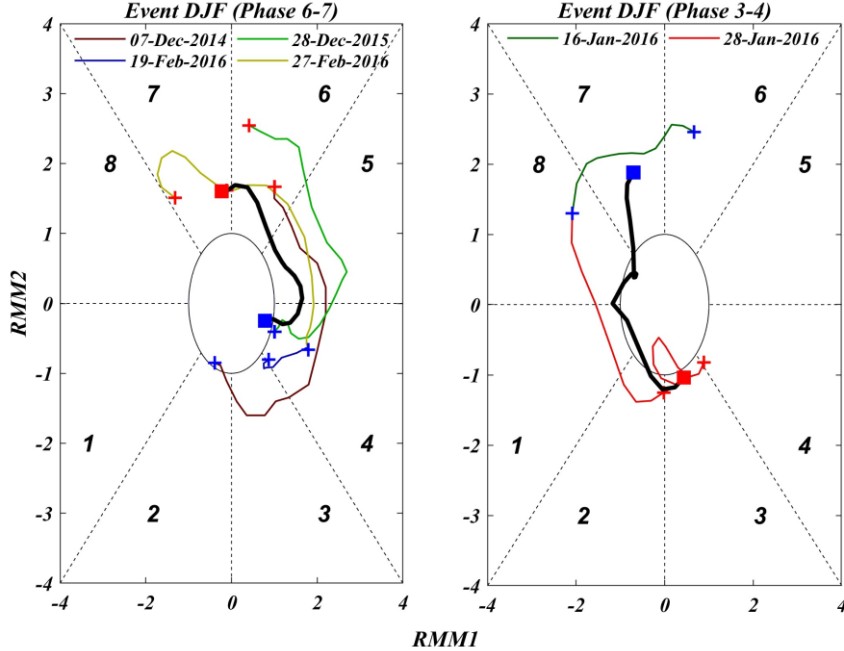


**Figure 9: Wheeler-Hendon MJO phase diagrams for 21-day-long trajectories preceding extreme events of high chl-a from the Puyuhuapi buoy (Table 3). Panel a show the subset of events peaking on MJO active phases 6-7, and panel b to the subset of events peaking on MJO active phases 3-4. The peak day (day 0) of each event is marked with a red cross, and the trajectory (colors curve) starts on day -20, denoted by a blue cross. The thick curve shows the mean trajectories considering all events, with similar colors for the beginning (blue square) and end (red square).**

Now, specifically regarding SLP, SST and wind speed anomalies, a complementary analysis is performed from the perspective
of MJO variability instead of focusing on extreme chl-a events. Such an analysis consists of investigating the composite
anomalous conditions prevailing during each active phase of the MJO for 1981-2010 using ERA5 data. For this, only active
days (MJO amplitude greater than 1) are considered. Figure 10 shows that phases 8, 1, 2, and 3 show a predominance of
negative SLP anomalies of up to -0.7 hPa (approximately 30% compared to the interannual variability anomalies, Fig. 2b)
throughout North and Central Patagonia. Inversely, phases 4, 5, 6 and 7 show positive SLP anomalies reaching up to 0.6 hPa
over the same region, these phases also show a predominance of south wind anomalies in the area. Phase 8 exhibits a strong
component of north wind anomalies. Positive SST anomalies around 0.3°C are observed throughout the Inner Sea of Chiloe
(around at approximately 46°S, 76°W) and off the Pacific coast during phases 8, 1, 2, and 6, with higher values in the first two
phases. Active phase 8 of MJO is similar to the day 0 composites of intraseasonal anomalies of the extreme events of chl-a
(Fig. 7), with negative SLP anomalies and on average north wind. These anomalies indicate that active phases 4, 5, 6 and 7
resemble (and consequently might reinforce) the average summer seasonal conditions showing relatively high SLP values over
the entire study area (Fig. 2a). In this way, average seasonal conditions tend to be reinforced (attenuated) during active MJO
phases 4, 5, 6 and 7 (8, 1, 2, and 3). In other words, we interpret that SY-IS anomalies during such active MJO phases might
superimpose constructively (destructively) with the mean summer condition.

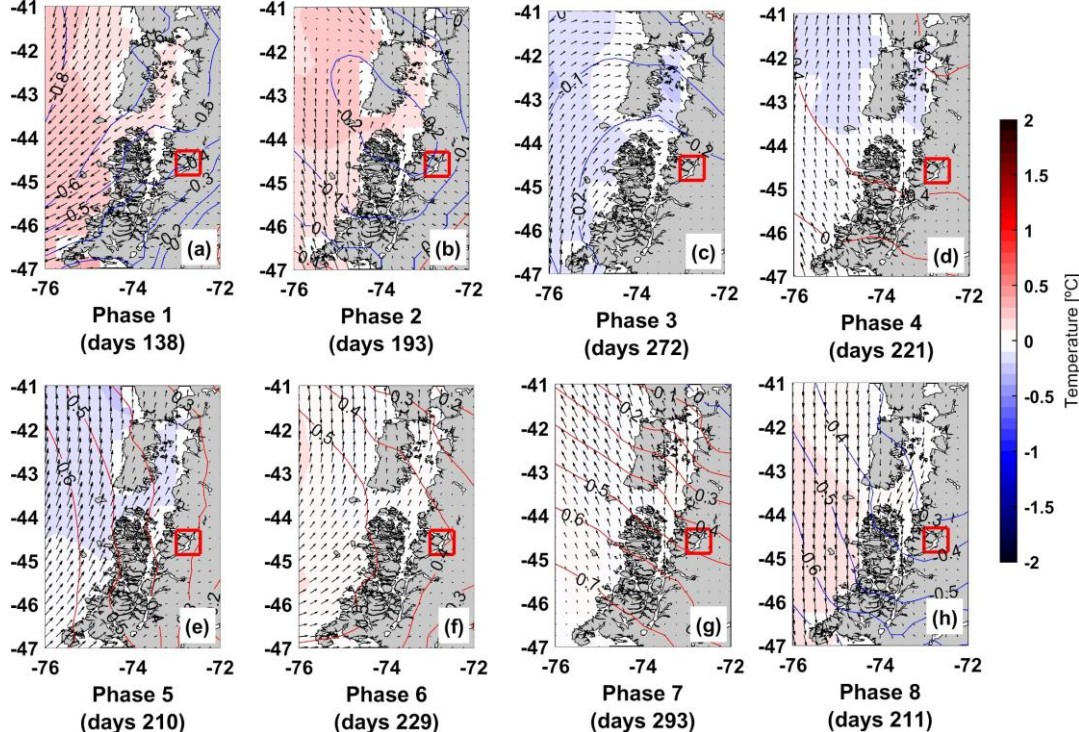


**Figure 10: Composites of intraseasonal anomalies for each MJO phase: SLP (contours; blue/red for negative/positive values), SST**
**(shaded), and wind direction and magnitude (vectors). Source: ERA5 Reanalysis.**
**4. Discussion**
The average climatic conditions of Chilean Patagonia Fjords during summer (DJF) are determined by the position of the South
Pacific Anticyclone, which is at its southernmost position during this season (Schneider et al., 2017 and Fig. 2). The associated
atmospheric circulation regime determines relatively lower precipitation and cloudiness and higher temperatures with respect
to winter (JJA). However, the influence of high-frequency climatic variability (synoptic-intra-seasonal, SY-IS) on the
Patagonian environment has received little attention so far. In this study we addressed this topic in relation to extreme events
of high summer chl-a concentration in the Puyuhuapi Fjord (PF) using daily in-situ data from an oceanographic buoy, CTD
monitoring station from Red Tide Management and Monitoring program carried out by Subpesca and IFOP, and reanalisys
data from ERA5.





In the PF, the seasonal summer conditions show a clear halocline and thermal stratification (Fig. 3), associated to a superficial
freshwater layer of 10 m that is relatively less saline and higher temperature due to the freshwater discharge and the direct
action of incident solar radiation respectively. The freshwater layer is deeper in summer than in winter due to a higher
freshwater runoff from precipitation and tributaries discharges (Schneider et al., 2014; Pinilla et al., 2020). Average chl-a
values are higher in JJA than in DJF due to favourable conditions for productivity in the water column: mainly in late JJA. In
addition, higher seasonal chl-a reaching ~3 µgL$^{-1}$ (Fig. 1b) are recorded at the northern and southern ends of the fjord (Jacaf
fjord and Moraleda Channel, respectively), which could be a result of the intrusion of oceanic waters richer in nutrients
(Schneider et al., 2014). On average, the water column at station A19 (Magdalena Sound), the closest to the Puyuhuapi buoy
(Bpuy) (44º35'17''S - 72º43'37''W), shows a SON-DJF stratification with an Haralambidou stratification parameter of 0.75,
stronger than at the other stations.
Considering DJF 2010-2018, 6 extreme chl-a events were found based on intensity and persistence criteria using the 90th
percentile (p90, 9.1 µgL$^{-1}$). The day of maximum anomalous chl-a values was defined as day 0. When analyzing the average
atmospheric and oceanographic conditions of the 10 days preceding each event, similar trends were observed for SLP, surface
temperature (T) and salinity (S) (Fig. 5). Specifically, decreasing trends in SLP and temperature were observed between days
-8 and -1. Salinity shows an increasing trend from approximately day -5 to day 5, as does Cisnes River streamflow between
days -7 and 0. This joint behavior of the variables may be indicative of upwelling of colder and nutrients rich subsurface waters
due to vertical mixing induced by the wind gyre. The tidal amplitude before and during each of the extreme events of high chl-
a shows no relation with the occurrence of the event, therefore most likely mixing by winds is more effective than tidal mixing
for the onset of these events (Fig. S6 in SM).
In DJF 2016, occurred 5 of the 6 extreme events recorded, including the one with the highest anomalous value (100 µgL$^{-1}$) on
19-Feb-2016 (Fig. 5a). That season has been the subject of several investigations (Buschmann et al., 2016; Garreaud, 2018;
León-Munõz et al., 2018) focused on major harmful algal blooms throughout the Chiloé Inland Sea area along with the
conspicuous accompanying drought and high radiation conditions. Therefore, our analysis focuses mainly on the months of
DJF 2016. Considering the average seasonal conditions, this summer was characterized by a stable atmospheric condition due
to anticyclonic anomalies over the southern tip of the continent and off the Pacific coast (Fig. 2b). DJF 2016 was characterized
by positive pressure anomalies south of the climatological position of the eastern South Pacific High. These caused a
weakening of westerly winds (40ºS - 50ºS), from which a decrease in the frequency of storms is inferred, and therefore a
decrease in precipitation in northwestern Patagonia (Garreaud, 2018). Thus, negative cloudiness anomalies and very dry
conditions were observed that season (León-Muñoz et al., 2018; Garreaud, 2018). Beyond this seasonal background, similar
synoptic conditions were found for most extreme events of high chl-a. The analysis of the synoptic patterns on the days of
their occurrence revealed the presence of strong negative SLP anomalies over the entire area, with low-pressure centers SW
of Patagonia indicating the passage of frontal systems around days -7 and 0 of each event (Fig. 2c).
Four of the 5 events corresponding to DJF 2016 show similar configurations in the SY-IS anomaly composites. In addition to
the presence of negative SLP anomalies, those events show a decreasing trend in T and an increasing trend in salinity between



days -10 and 0 (Fig. 6a-c). Negative SLP anomalies are also evident in the composite sequence between days -7 and 0, along
with a cyclonic wind shift between days -7 and 0 (Fig. 7). This suggests that the passage of this low-pressure system causes
an increase in vertical mixing as a result of wind gyre, that promotes colder subsurface waters with higher salinity values in
surface, inducing a weakening of the haline stratification. The mixing induced by the wind gyre drags water and causes a
weakening of the thermal stratification. Enhanced cloudiness persists for about 7 days; SST decreases due to reduced direct
solar radiation and the ascent of colder water to the surface.
The analysis of high-frequency anomalies allows us to propose a mechanism for occurrence of extreme events of high chl-a
which the passage of a low-pressure system seems to favor conditions for high chl-a values in the PF. In this sense, the
mechanism suggested in this study supports the results of Montero et al. 2011 and Collins et al. 2019, which conclude that
productivity patterns may be associated with mesoscale changes in the wind pattern. Specifically, they suggested that wind
forcing induces modifications in water column stratification during winter in the PF (Montero et al. 2011), and during spring
in the Georgia Strait area (British Columbia, Canada; Collins et al. 2019). Considering the above, it is necessary to clarify that
there are different atmospheric and oceanographic mechanisms, which vary according to the geographical area and the season,
and provide conditions for the occurrence of an extreme event of high chl-a. One case, is the study by Jacques et al. (2023),
where the atmospheric conditions were totally different to those indicated for FP, with the passage of a high pressure system
(migratory anticyclone) that favoured the conditions for an increase in phytoplankton biomass.
In order to explore the potential linearity of the relationship between atmospheric conditions and chl-a concentration in Bpuy,
we analyzed extreme events of low summer chl-a anomalies, using the 10th percentile (p10, -24 µgL$^{-1}$) as the intensity
threshold and a criterion of persistence. Specifically, Figure 8 exhibits positive SLP anomalies between days -3 and 0, an
increasing trend in T between days -7 and 0, and a decreasing trend in S from day -5. The signal from the streamflow anomalies
of the Cisnes River was not taken into account due to the absence of data at that time. In particular, the case study corresponding
to 28-Dec-2016 shows an increasing (decreasing) trend in SLP and SST (salinity, S) (Fig. 8), indicating that these anomalies
constrain a favorable environment for chl-a events. As expected from mid-latitude synoptic systems, the increase in SLP
anomalies (tendency to anticyclonic conditions) is associated with decreased cloudiness, increased incoming solar radiation
and warming of the sea surface. In general, the characteristics of these events were inverse to those found in the positive events,
suggesting a possible linear relationship between environmental forcing and chl-a response. This strengthens the mechanism
proposed for the occurrence of extreme events of high chl-a, because, under these conditions of low chl-a events, stratification
is reinforced, limiting nutrient availability, which corresponds to one of the potential triggers of chlorophyll episodes in the
fjords.
Interestingly, all 6 extreme events of high chl-a in DJF peak during active MJO conditions (Table 3). This is an indication that,
active signal of the MJO may induce favorable conditions for the occurrence of high extreme events of chl-a anomalies through
teleconnections (Jacques-Coper et al., 2023; Gomez et al., 2017). Phases 6 and 7 are the predominant phases in the 2016 events,
with two events in each of these, which may be an indicator that these phases could modulate the hydrographic environment
of the PF and favor the occurrence of chl-a events.




463 Complementarily, we analyzed the modulation of SLP, SST and wind anomalies induced by all active MJO phases (Fig. 10).

464 Phases 6 and 7 indicate positive SLP anomalies over the entire study area. This result suggests that these active MJO phases

465 might reinforce, in the SY-IS timescale, the mean seasonal atmospheric conditions observed in DJF 2016. But, the results of

466 the present study indicate that the conditions associated with SY-IS variability would be of opposite sign to SY-IS anomalies

467 during phases 6 and 7. However, it is necessary to be attentive when the oscillation is in phase 6 or 7 active that can cause an

468 increase in chl-a, because it can present atypical conditions in some cases, different from the average conditions in the study

469 area, which may reinforce SLP anomalies due to the passage of a low-pressure system.

470 The conceptual framework of the present study is based on the superposition of anomalies from different time scales,

471 specifically the annual cycle, the seasonal anomaly, and the synoptic e intraseasonal anomalies evolving towards the day of

472 maximum event. In this context, the following discussion on the contrast between SY-IS variability and the other components

473 is necessary. Over the characteristic range of variability associated with the MJO, phenomena of higher frequency can be

474 superimposed on the subsynoptic and synoptic scale (from hours to days). These phenomena, in this case the passage of a low-

475 pressure system, with a duration of less than 7 days, seem to be a relevant forcing for the extreme chl-a events analyzed in this

476 research. In this sense, our mechanism suggests that these synoptic phenomena are mainly responsible for favoring conditions

477 for the occurrence of extreme events of high chl-a. Accordingly, the result could improve predictions of the occurrence of

478 extreme high chl-a events, due to the high level of predictability of these synoptic phenomena that generate the optimal

479 conditions for the occurrence of events of high values of chl-a.

480 Therefore, on the seasonal and interannual variability of the summer of 2016 and the anomalies produced by the MJO, which

481 impose conditions of strong stratification and light availability, the synoptic-intraseasonal variability provides the necessary

482 nutrients for the occurrence of an extreme algal bloom event.

## 5. Conclusions and perspectives

484 The main conclusion of this research is based on the proposition of a high-frequency variability mechanism that generates

485 favorable conditions for high chlorophyll in the summer season. Specifically, we suggest that positive SLP and radiation

486 anomalies, and a characteristic summer vertical stratification in the water column are observed before the occurrence of these

487 events (days -10 to -7); thereafter (days -7 to 0), the passage of a low atmospheric pressure system is observed, causing a

488 change in wind direction and intensity, and a decrease in temperature, which in turn causes strong vertical mixing of the water

489 column, weakening the stratification causing to increased nutrient availability near the surface. In summary, we describe that

490 atmospheric processes show the ability to modify the hydrographic conditions of the Puyuhuapi fjord during summer. Our

491 results motivate future research that might arise based on longer time series and more frequent observations, considering also

492 other seasons of the year, in order to gain a deeper understanding of the modulation of SY-IS on hydrographic environments,

493 given the relevance of these phenomena and the influence they have on the processes in the fjords.



Improving the predictive ability of algal blooms and their relationship with climate conditions from intraseasonal to interannual scales is an ongoing process that requires a multidisciplinary approach, data-driven analysis, and a commitment to refining and enhancing predictive models over time. Thus our finding regarding empirical relationships between MJO or other intraseasonal processes and algal blooms contributes to develop early warning systems that can provide timely alerts to authorities, stakeholders, and the public when conditions are conducive to algal blooms.

**6. Competing interests**

The contact author has declared that none of the authors has any competing interests.

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
