# Peer review of "Synoptic-intraseasonal variability control on high chlorophyll-a"

_EGUsphere, 2023_

## Author Comment (AC1)

Responses to reviewer 1

We thank reviewer 1 for her/his review. We have considered all the comments and revised the manuscript accordingly. Our detailed responses are listed below.

**1. The number of extreme chl-a events in the study is too low to base any reliable statistical conclusion on. Also, it is not clear how many events there actually are. In the abstract, it is stated that 6 (83%) of the extreme events occur in 2016. This would imply 7 events. However, in table 4, there are only 6 events listed in total (with 3 of these in (December) 2016). So it is not clear how many events there are, but either 6 or 7 is far too small a number to base any reliable conclusions on.**

Thanks for this point. In the analysed period 2010-2018, 6 extreme events of high chl-a values were found, and 5 of them (83%) correspond to 2016 (Table 3). In fact, there was an error in the abstract, and in the revised version of our work we replaced the original sentence:

*'Six extreme events, corresponding to 83% of the total, were detected in the year 2016 ...'*,

with the following one:

*'Out of six extreme events, five (83%) were detected in the year 2016…'*.

We clarify that the main objective of our study is to analyse whether the synoptic-to-intraseasonal (SY-IS) environmental variability is related to the occurrence of extreme events of high chl-a in the Puyuhuapi fjord located in Chilean Patagonia (the six events listed in Table 3). In this context, extreme events of low chl-a (the six events listed in Table 4) are also included to explore whether any signs of linearity are evident between chl-a and the prevailing atmospheric conditions. This is indeed the case, as revealed in Figure 5 and discussed in subsection 3.3.

We agree that we are dealing with a very limited number of events, but this is a challenge to face in this kind of studies. The in-situ time series we analyse is unique in its nature in the context of the Chilean fjords, a fact that makes it particularly valuable for this kind of research. In most of these fjords, very scarce sampling is carried out, with a frequency up to two values per month. By definition, the extreme events we analyse correspond to values within the highest 10% SY-IS anomalies between 2010-2018 (as defined in the methodology). As shown in Fig. 6, they comprise well-defined events that show evident deviations from the long-term mean. Given the length of the available observations (2010-2018) and the definition of these extreme events according to both intensity (exceedance of the 90th percentile) and persistence (of at least 2 days), we agree that it is not possible to compute robust statistics based on these extreme events. However, we can still investigate their characteristics and analyse whether there is any similarity in the conditions before and during their occurrence, and thus establish a characteristic atmospheric configuration that is recurrent in most of the events and also suggest an associated physical mechanism. These are all valuable goals on their own. However, we agree that it is crucial to clarify to the readers the inherent restriction of our study framework. To underline this aspect of our work, we have included the following sentence to the conclusion section of the revised version of the manuscript:

*'We are aware that no robust statistical conclusions can be drawn from the limited number of extreme events of chl-a that arise from the definition applied on the limited length of observations analysed here. However, our results contribute to establish an analysis framework and deliver indications for atmospheric monitoring. Therefore, we encourage revisiting our results when observations spanning a longer period become available.'*

**2. Of the 6 high chl-a events listed in Table 4, the MJO phases are also listed. These are, for the 6 events, MJO phases 4,6,7,2,6,5. In the abstract, it is concluded that MJO phases 6-7 tend to reinforce chl-a events. This is an incorrect conclusion. 3 out of 6 of the high chl-a events occur in MJO phases 6-7. With such a small number of events, and no prior reason to think that MJO phases 6-7 re important, this can easily happen by pure chance. Also, it is stated that ENSO is an important modulator of the high chl-a events, as 2016 is an ENSO year. The paper only analyses DJF data, and it is not clear whether the 2015/16 or 2016/17 DJF season is being discussed, which is very confusing. What is the state of ENSO in these two DJF seasons. The whole discussion here is confusing and not backed up reliably by the data.**

As clarified in the previous answer, Table 4 exhibits the extreme events of low chl-a. The MJO phases registered during the extreme events of high chl-a are shown in Table 3. In this latter group of high chl-a, 4 out of 6 (67%) events correspond to active MJO phases 6/7. Building upon this fact, we analysed the corresponding trajectories (between days -20 and 0) in the Wheeler-Hendon phase diagram for each of these events (Fig. 9a; Fig. 9b exhibits the events peaking on active MJO phases 3 and 4). We found that all 4 events had similar trajectories (however, it must be noted that the events peaking on 19-Feb-2016 and 26-Feb-2016 overlap with a span of 8 days). All these trajectories developed during active MJO phases (in particular phases 4 and 5) prior to the occurrence of their associated extreme events. This result seemed as an indication for a possible relationship between the active MJO phases 6/7 and favorable atmospheric conditions for the build-up and triggering of the extreme events. From our results, there is no evidence that other active MJO phases (e.g. 1,2,8) are associated with similar conditions. Moreover, no chl-a extreme events were found during inactive MJO conditions (i.e., intensity of the Wheeler and Hendon index below 1).

To further explore this, and as a complementary analysis from the "MJO perspective" (regardless of the extreme events), we computed the mean fields of intraseasonal anomalies in the study area for each MJO active phase, which are shown in Fig. 10. We found that positive SLP anomalies and a southerly wind component predominate mainly in these active phases 6 and 7. Therefore, this suggests that active MJO phases 6/7 could reinforce, in the SY-IS time scale, the seasonal mean atmospheric conditions observed in DJF 2016 (previously shown in Fig. 2b). Taking this possible constructive interaction into account, we infer that the MJO-related anomalies seem to establish a favorable scenario for the occurrence of extreme events of high chl-a.

Correspondingly, the following comment was reformulated in the abstract:

*(…) Furthermore, this work suggests that active phases 6 and 7 of the MJO might reinforce, on the SY-IS timescale, the seasonal atmospheric circulation observed in DJF 2016 (December/2015-February/2016). (…)*

With respect to 2016: previous research has shown the influence of large-scale climate modes, particularly ENSO and the Southern Annular Mode (SAM), on the inter-annual variability of phytoplankton blooms in the spring-summer seasons (September/October/November-December/January/February) (Lara et al., 2016). The summer of 2016 was unusual in that a harmful algal bloom of *Pseudochattonela cf. verruculosa* wiped out about 12% of Chilean salmon production, causing the worst mass mortality of fish and shellfish ever recorded in the coastal waters of western Chilean Patagonia. This bloom occurred during a strong El Niño event and the positive phase of the Southern Annular Mode (SAM) that caused anomalies in the atmospheric circulation in southern South America and the adjacent Pacific Ocean that in turn caused dry conditions with positive radiation anomalies during those months (León-Muñoz et al., 2018).

Considering that these modes of variability, particularly ENSO, have their spectral peak in the interannual scale, they lie beyond the main focus of our research. Recall that the interannual signal was removed from our data through equation 1. Indeed, our main goal is to explore periods shorter than the interannual scale. The revised text highlights this.

Following previous ideas, the introduction was reformulated to provide a more complete context to the reader and to account for the importance of inter-annual modes of variability scale in the modulation of phytoplankton blooms, specifically in the summer of 2016. Now, this section comprises a passage that reads:

*Previous research has shown the influence of large-scale climate modes acting in the inter-annual scale, such as El Niño-Southern Oscillation (ENSO) and Southern Annular Mode (SAM), on the intensity of phytoplankton biomass in the spring-summer seasons (SON-DJF) (Lara et al., 2016; Garreaud, 2018; León-Muñoz et al., 2018). The most salient recorded event occurred in DJF 2016 (December/2015-February/2016), during which a harmful algal bloom of Pseudochattonella cf. verruculosa wiped out about 12% of Chilean salmon production, causing the worst mass mortality of fish and shellfish ever recorded in the coastal waters of western Chilean Patagonia. This HAB took place during a strong El Niño event and the positive phase of SAM that induced anomalies in the atmospheric circulation in southern South America and the adjacent Pacific Ocean, which in turn caused dry conditions with positive radiation anomalies during those months (León-Muñoz et al., 2018; Garreaud, 2018).*

**Further relevant changes in the revised version of the manuscript**

In line with our previous answers, we incorporated some discussions and clarifications in the revised version of the manuscript, with the aim of clarifying some of the ideas set out in the article that were not explicitly or obviously described but were considered in the research.

Original text in normal font; new text in italics

1 Introduction

47    '(...) From a climatic point of view, SY-IS variability represents high frequencies that break into the seasonal variability generating a wide range of responses on oceanographic and biological variables.the seasonal variability generating a wide range of responses on oceanographic and biological variables'.

*'(...) From a climatic point of view, SY-IS variability represents high frequencies that break into the seasonal variability generating a wide range of responses on oceanographic and biological variables. SY-IS might arise on top of seasonal variability, and both might superimpose constructively or destructively.'*

66    'Jacques-Coper et al. (2023) found that high biomass events in Inner Sea of Chiloé occurred under the influence of a mid-latitude migratory anticyclone, inducing negative cloudiness (or increased photosynthetically active radiation: PAR) and positive SST anomalies.'

*'Jacques-Coper et al. (2023) found that high biomass events in Inner Sea of Chiloé occurred under the influence of a mid-latitude migratory anticyclone, inducing negative cloud cover anomalies leading to positive anomalies of photosynthetically active radiation (PAR) and SST. This association between atmospheric conditions and events of high chl-a in the Puyuhuapi Fjord may help to identify the synoptic configuration that may tend to favor them, by possibly forcing the marine environment. Moreover, if such configuration is modulated by climate variability modes, guidance on the predictability of their occurrence might be revealed. In any case, it should be kept in mind that SY-IS climate variability might just be one factor influencing the increase of phytoplankton biomass. Many other factors could be of relevance, for example, the availability of inorganic nutrients (from land-based activities through runoff) and the trophic interactions of the organisms present in the water column.'*

4 Discussion

420    '(...)These caused a weakening of westerly winds (40ºS - 50ºS), from which a decrease in the frequency of storms is inferred, and therefore a decrease in precipitation in northwestern Patagonia (Garreaud, 2018).'

*'(...)These caused a weakening of westerly winds (40ºS - 50ºS), from which a decrease in the frequency of storms is inferred, and therefore a decrease in precipitation in northwestern Patagonia (Garreaud, 2018). These climate anomalies were related to a strong El Niño (ENSO) event and the positive phase of the Southern Annular Mode (SAM) (León-Muñoz et al., 2018; Garreaud, 2018). Considering that these mentioned modes of variability, mainly ENSO, belong to the interannual scale, they are not within the scope of our research, because the interannual signal was removed of our data, so no results of these are included.'*

460    '(…)Phases 6 and 7 are the predominant phases in the 2016 events, with two events in each of these, which may be an indicator that these phases could modulate the hydrographic environment of the PF and favor the occurrence of chl-a events.'

*'(...)MJO phases 6 and 7 are the predominant phases in the 2016 events, with two events each. This may be an indicator that these phases could modulate the hydrographic environment of the PF and favor the occurrence of extreme events of high chl-a. On the other hand, 4 out of 6 (i.e., 67%) of the extreme events of low chl-a occur during non-active phases of the MJO. Of such extreme events of low chl-a, only one occurs during an active phase 6. This reinforces such a possible association.'*

*'Complementarily, we analyzed the modulation of SLP, SST and wind anomalies induced by all active MJO phases (Fig. 10). Phases 6 and 7 indicate positive SLP anomalies over the entire study area, leading to negative cloud cover anomalies and an increase in PAR. This result suggests that these active MJO phases might reinforce, in the SY-IS timescale, the mean seasonal atmospheric conditions observed in DJF 2016 (Fig. 2b). Consequently, we infer that the background condition established by the annual cycle can be modified (reinforeced or weakend) by the intraseasonal variability., However, in this case, we find that the synoptic variability imposes a crucial determinant factor that can be very relevant for the occurrence of an extreme event of high chl-a.'*

**5. Conclusions and perspectives**

**490**     '(...)Our results motivate future research that might arise based on longer time series and more frequent observations, considering also other seasons of the year, in order to gain a deeper understanding of the modulation of SY-IS on hydrographic environments, given the relevance of these phenomena and the influence they have on the processes in the fjords.'

*'We are aware that no robust statistical conclusions can be drawn from the limited length of the observations analysed here and the definition of the extreme events of chl-a. Therefore, we encourage revisiting our results when observations spanning a longer period become available. Our results motivate future research that might arise based on longer time series and more frequent observations. Future studies could also consider other seasons of the year, in order to gain a deeper understanding of the modulation of atmospheric SY-IS variability on hydrographic environments, given the relevance of these phenomena and the influence they have on the processes in the fjords. '*

---

## Author Comment (AC2)

**Responses to reviewer 2**

We thank reviewer 2 for her/his review. We have taken all the comments into account for a revised version of our work. The list of our detailed responses can be found below.

**-The Introduction should be more straightforward and introduce the reader to your objectives, following an easily understandable order. The section starts describing the seasonal drivers of phytoplankton for the Chilean Patagonia and then jumps to the Maden-Julian Oscillation (MJO) without first mentioning why it is doing all this and what is the relevance of studying phytoplankton and, more precisely, what is the relevance of studying it in a fjord in Patagonia. It is important that the overall public that reads the article understands how globally relevant it is. Moreover, if the main goal of the study is to understand how intraseasonal variability drives phytoplankton biomass, why not focus more on phytoplankton and then gradually move to how intraseasonal climate variability may affect it? This is more a matter of restructuring the Introduction to make it more appealing for the reader.**

Thanks. We agree on this point. The introduction will be restructured with the aim of guiding the reader and making the text more attractive, highlighting the relevance of this study, and how innovative it is. Our main aim is to study and describe the possible association between events of high biological activity in a Patagonian fjord and particular atmospheric configurations through ocean-atmosphere interactions. Furthermore, we aim at investigating whether such atmospheric configurations are modulated by modes of climate variability. Our results indicate that this is indeed the case for extreme events of high chl-a in the Puyuhuapi fjord, which reveal a teleconnection with the intra-seasonal Madden-Julian Oscillation (rooted in the tropics) and also with synoptic variability. These results contribute to enhance the predictability of said events.

**-The objectives are not clear. "The research aims to suggest a mechanism by which SY-IS variability modulated water column conditions, triggering extreme chl-a events". Shouldn't it be the other way? For instance, the goal should be to first understand what is driving these events, then find out the main frequency of these environmental factors and finally what are the overall process? Yet, in the beginning of the paragraph (line 70), the authors mention that the study will also focus on seasonal variability. Finally, I do not understand lines 75-77, it is not clear if this still part of the objectives or not or what is the clear relationship between it and the extreme chl-a events. I suggest rethinking the objectives – keep it simple, avoid repeating objectives and using vague terms that the reader is not familiar with at this point text.**

In fact, the main objective is to explore whether the occurrence of extreme events of high chl-a are associated with certain atmospheric configurations in the synoptic-intraseasonal (SY-IS) variability range. Then, specific objectives are 1) to analyse the role of the SY-IS variability range behind the occurrence of such extreme events and their possible superposition on anomalies that stem from variability at longer timescales (seasonal and interannual), 2) to investigate potential atmospheric configurations associated with extreme events of low chl-a, in order to assess the possible causality and linearity of the atmosphere/chl-a interaction, 3) to establish whether modes of SY-IS climate variability seem to modulate the occurrence of said extreme events.

It is important to highlight that the framework of our study is based on the fact that an actual atmospheric configuration might be conceptualised as the superposition of anomalies

characteristic of different time scales, namely the mean annual cycle, the seasonal anomaly (corresponding to the season-to-season change, i.e. interannual variability) and the SY-IS anomalies. Some of these components might evolve constructively together towards conditions favouring the extreme events. This is why we describe how the seasonal conditions (belonging to the mean annual cycle) modulate some of the limiting factors for productivity in the fjord, then the interannual anomalies for selected summer seasons, given that some of them may present strengthened or weakened seasonal conditions in comparison to the mean values, and, on top of these, we focus on SY-IS-scale phenomena, which –as shown by our results– are crucial for establishing the conditions for the occurrence of extreme events of high chl-a. Such approach is part of the novelty of our contribution for this kind of studies, since traditional approaches tend to focus mainly on the interannual variability.

**The Methods section has several major problems. First, the section lacks several details and is confusing, which makes it difficult to follow at times and, more importantly, difficult to reproduce. I would suggest adding a summary table with all variables used, with their sources and resolution. I have also indicated several instances where the methods could be improved below, after my general comments.**

Thanks. The revised version of our manuscript will incorporate these suggestions to clarify the methods used in our study. In particular, we will include the summary table with the specific information.

**Chl-a is the main variable of this study. Yet, all results are based on in-vivo fluorometry chl-a measured on a buoy within a highly productive and potentially turbid fjord. Without validating these measurements against lab-measured chl-a (HPLC, preferentially), it is difficult to be sure that these results are reliable as currently shown. Particularly, when chlorophyll-a concentrations above 100 mg/m3 are common from what I can see in Figure 4. In-vivo fluorometry are often less accurate and frequently overestimate chl-a (over 2-fold) due to the fluorescence of coloured dissolved organic matter. Other matters such as turbidity, biofouling, and nonphotochemical quenching can also interfere with the measurements of chl-a. Note that many in-vivo fluorometers are not prepared to handle very high concentrations. Therefore, it is very important to ensure that your measurements are accurate, particularly when the entire goal of the work is evaluating the drivers of extreme events of chl-a. Personally, I have doubts that the values presented are accurate, which might affect the validity of the results.**

Thanks for your comments. This in-situ time series of Chilean fjords is unique in nature, which makes it very valuable for this type of research. Most of these fjords are very sparsely sampled, with a mean frequency of one measurement per month. The values measured by the chl-a sensor of the buoy used in this study were validated by Dr. Iván Ernesto Pérez-Santos in collaboration with Dr. Paulina Montero, by in-situ sampling and chl-a validation in the laboratory. These water samples for chl-a were filtered through 25 mm 0.7 μm GF/F filters and immediately frozen at −80 °C and extractions were performed in 10 mL of acetone 90% for 24 h at 4 °C in the dark. Chl-a was measured by fluorometry in a Turner Designs Trilogy Laboratory Fluorometer. The very high correlation between the in-situ and laboratory data (r=0.98, figure not shown) indicates that the buoy data are reliable. Furthermore, the overall maximum values of chl-a measured by the sensor

(exceeding 100 μg/L) correspond to the winter season, and these values are not used in our study. The values considered in our research corresponding to summer (DJF) and are below 65 μg/L.

**I also did not understand the choices made in the Methodology. For instance, if you have a rich dataset with summer and winter samples scattered along the fjord, why average them for the entire seasons? First, by doing this the authors are "throwing away" the intraseasonal information (the one which is the focus of the work) in order to have a single value for three months. Second, this makes it difficult for the reader to understand when these samples were collected and if they were equally collected along each season. I think the authors may be wasting the potential of their data as it is not clear why they only use intraseasonal chl-a data from the buoy and for looking at the extreme events.**

Thanks. The answer to this question is related with our second answer, as it has to do with the superposition of variability at different time scales. As indicated in the manuscript, such observations along the fjord correspond to measurements taken at sampling stations in different seasons of the year during the entire study period. These samples have a mean frequency of ~1 per month; in some sampling stations they were taken less frequently and in some they were taken twice a month. Therefore, w e cannot compute SY-IS anomalies from these values, because they do not correspond to daily data. Moreover, since particular measurements could correspond by chance to days with particularly high or low productivity, a single value might not be representative of the mean monthly conditions. For this reason, data from sampling stations were used only to show the average seasonal conditions (long-term mean) in the study area. The sampling distribution is shown in Table 1 and Figure 3 (in the original manuscript). Finally, it is important to clarify that not these station samples but the daily buoy chl-a data are used for computing the SY-IS variability of chl-a.

**I would really advise against using a product with such low resolution such as ERA5 for studying such a small, coastal, and complex region as a fjord. Not only it may not represent the full picture, but I am sure most pixels would also include land.**

Thanks for this point. We use the state-of-the-art ERA5 Reanalysis to characterize the synoptic and regional atmospheric variability around the study area, which is the highest atmospheric resolution targeted in our study. We do not pretend to use it to show the very specific conditions within the fjord, as much higher resolution would be needed indeed to resolve the corresponding phenomena. Nevertheless, the regional atmospheric conditions identified through ERA5 data before and during the occurrence of extreme events of high chl-a are sufficiently detailed to characterize the anomalies of the main atmospheric variables considered in this study and to define the synoptic patterns associated with the occurrence of extreme events of high chl-a.

**In section 2.4, why are the authors using these "intraseasonal anomalies"? This must be clear in the text. Also, does not this equation remove the same information twice? For instance, first it**

removes the average value for that day (let us say January 1st) then removes the seasonal anomaly (which already includes the value from January 1st). I may be misunderstanding the equation, but I do not believe this would be the best way to remove the seasonal and interannual variability to your datapoints, which is what it seems the authors are trying to do here. Moreover, since the authors are calculating anomalies from buoy chl-a, I would be really careful with the outliers prior to calculating the anomalies. Finally, I am not sure I agree with the approach of defining day 0 as the maximum day of the event and analysing the conditions prior and after it. There are several environmental factors that can lead to the abrupt ending of a bloom and there is often a lag between the environmental change and their effect on phytoplankton biomass. Plus, a bloom in December can be completely different from a bloom in February, both in terms of community composition, nutrient availability, and grazing influence (all factors not included in this work). Overall, this analysis might be too simplistic for the authors' goal.

Thanks for sharing these concerns. There are in fact many alternative ways for the calculation of "pure" SY-IS variability. We use equation (1) to calculate the SY-IS anomalies in summer (DJF), which has been successfully applied in previous investigations addressing the SY-IS timescale, as shown in the references of our manuscript. The SY-IS anomaly correspond to the residual that results after subtracting from the daily value its corresponding daily climatological mean (in this first step we eliminate the expected values and their annual cycle), then the difference between the particular seasonal mean of that specific year and the long-term seasonal mean of the whole period (in this case, DJF 2010-2018, thus eliminating the interannual [i.e., season-to-season] variability). There is no repeated subtraction of the same term. As explained in the previous answer, the highest values of chl-a (those above μg/L) are not considered in this study when calculating the intra-seasonal anomalies, because these outliers are mostly found in winter months (Figure 4 in the original manuscript).

In addition, the choice of defining "day 0" as the day of maximum chl-a anomaly in each extreme event is based on the fact that this is an evident and representative aspect, common to all events. Taking into account the lag between SY-IS environmental changes and their effect on phytoplankton biomass, our analysis focuses on the conditions prior to day 0 in a time span of 10 days. Following this procedure, our results reveal persistent anomalies from day -6 onwards in the composite evolution, exhibiting that a transient mid-latitude low-pressure system is key to the generation of the maximum chl-a values that define the extreme events. From these results we infer that these persistent atmospheric conditions trigger an extreme event around day 0. In any case, it should be noted that we are aware that synoptic atmospheric variability is just one of the factors influencing the evolution of phytoplankton biomass. Our study aims at contributing to the understanding of this phenomenon and future studies could build upon its results as more high-frequency observations of a wide range of variables become available for the study area.

**In the results, why are we looking at the extreme events in the summer if (according to Figure 4) the highest biomass are observed in other periods of the year (autumn and winter)?**

We focus just on extreme events occurring in summer because of the impacts these blooms have historically had on the ecosystem, human health, and aquaculture. Since some of these summer blooms were associated with toxic species (e.g. *Alexandrium catenella, Pseudochatonella*

*verruculosa*) and have spread for hundreds of kilometers, they have induced massive fish mortality and shellfish contamination, inducing great concern due to their large impacts in Chilean Patagonia.

**What is the rationale behind the "extreme low chl-a" events? Why are these events even relevant for the objectives of this study?**

Thanks for raising this point, which is crucial for our study. Please refer also to our second answer. The study of extreme events of low chl-a helps us to understand whether there is any degree of linear relationship between extreme chl-a values and atmospheric forcing, and raise a further indication for a potential causality between them. In fact, our results reveal that opposite extreme events of chl-a (i.e. high and low chl-a) tend to exhibit also relatively opposite atmospheric configurations. This aspect will be better described in the revised version of our manuscript.